# Oral Health Behaviour of Nine-Year-Old Children and Their Parents in Sarajevo

**DOI:** 10.3390/ijerph18063235

**Published:** 2021-03-21

**Authors:** Enes Karamehmedovic, Elmedin Bajric, Jorma I. Virtanen

**Affiliations:** 1Department of Clinical Dentistry, University of Bergen, 5009 Bergen, Norway; enes.k401@gmail.com; 2Department for Preventive Dentistry and Pedodontics, Faculty of Dentistry with Clinics, University of Sarajevo, 71000 Sarajevo, Bosnia and Herzegovina; elmedinbajric@gmail.com; 3Institute of Dentistry, University of Turku, 20520 Turku, Finland

**Keywords:** children, knowledge, oral health behaviour, parent-child link, dental health

## Abstract

The oral health situation in Bosnia and Herzegovina is among the worst in Europe. We investigated the oral health behaviour of primary schoolchildren and their parents in Sarajevo. This was an anonymous cross-sectional survey among third-grade schoolchildren and their parents’ oral health habits in Canton Sarajevo. Cluster random sampling yielded a representative sample from all the public schools in Canton Sarajevo in 2019. The survey targeted a total of 441 children and 365 parents. Two thirds (66.5%) of the children reported brushing their teeth twice daily, and almost half of them failed to use fluoride toothpaste daily. Girls brushed their teeth significantly more often than did the boys (74% vs. 58%, *p* = 0.004). Children living in residential areas of middle and high socioeconomic status (SES) reported better oral health habits than did those living in areas of low SES. Our study showed that Sarajevo children’s oral health habits were poor. One-third of the nine-year-olds failed to brush their teeth according to recommendations, and almost half of them failed to use fluoride toothpaste daily. Improving the children’s oral health in the future will urgently require national oral health promotion and prevention programmes.

## 1. Introduction

Dental caries is the most widespread non-communicable disease and a major public health problem globally. Oral health inequalities exist among and between different population groups around the world [1]. Especially susceptible to these inequalities are children in low- and middle-income countries as well as in socially marginalised areas and communities [2]. Social determinants strongly impact oral health. Poor oral health habits and consumption of free sugars are widely recognised as primary risk factors for oral diseases and the World Health Organization (WHO) has advocated for prevention [3,4,5]. Both poor oral health habits and high sugar consumption often occur in the same individual [6] with adverse cumulative health effects that worsen from childhood to adulthood.

The parents’ role in instilling in their children a positive attitude towards and promoting their children’s oral health behaviour at home is crucial [7]. Parental beliefs about the tooth brushing behaviour of their children matter [8]. Parents serve as important daily role models for their children, so their oral health behaviour and oral health status correlate significantly [8,9]. Parents should instil in their children good health attitudes and behaviour. The children of parents who brush according to recommendations and those who take control of the children’s oral health habits have better oral health status [10,11].

Regular tooth brushing with fluoride toothpaste is linked to better oral health status and significantly reduces dental caries. Lack of regular and proper oral hygiene leads to a higher risk for caries [12]; dental professionals broadly agree that regular oral health behaviour starts with the eruption of the first teeth [13].

Central and Eastern European countries have shown a higher prevalence of dental caries among children; in Bosnia and Herzegovina, for instance, the risk for dental caries is twice the regional average among 12-year-olds [14]. Overall, the oral health situation in Bosnia and Herzegovina is among the worst in Europe [15]. The DMFT (the number of Decayed teeth, Missing teeth due to caries and Filled teeth) values are high: the DMFT indices in Bosnia and Herzegovina among 12–15-year-olds was 7.2 and among five- to seven-year-olds 7.5 [16]. The corresponding scores in other European countries have been 0.9–7.7 and 0.9–5.4, respectively [16,17]. Development has been slow [18]. The latest national reports highlight inferior oral health with DMFT indices of 7.2 among 6- to 12-year-olds and of 4.9 among 3- to 5-year-olds [19].

“Bosnia and Herzegovina is a region with low levels of natural fluoride (less than 0.1 ppm) in the drinking water” and no water fluoridation [20]. Fluoride toothpaste has been available on the market for many years, but research on the frequency of their use as a share of a population generally reluctant to accept the importance of fluoride is lacking [21].

A family’s socio-economic status (SES), oral health knowledge, attitude and behaviour are considered risk factors for dental caries [7]. Parental education is a significant part of SES [3,22]. The literature indicates that SES critically influences dental caries with a clear correlation between low SES and children’s oral health: children in low SES groups have higher values for decayed and filled primary dental surfaces [2] and a higher prevalence of dental caries [3], which is considered a risk factor for poor oral health knowledge [23].

This study aimed to investigate the oral health behaviour of elementary schoolchildren and their parents in Bosnia and Herzegovina. Our hypothesis predicted that parents with better oral health behaviour and higher oral health knowledge who live in better socio-economic conditions would have children with better oral health habits.

## 2. Subjects and Methods

This study was a cross-sectional survey in primary schools among third-grade schoolchildren and their parents in Canton Sarajevo, the largest administrative centre in the Federation of Bosnia and Herzegovina. Cluster random sampling yielded a representative sample from all the public schools in Canton Sarajevo in 2019.

### 2.1. Subjects

Canton Sarajevo has a total of 78 primary schools in its nine regions/municipalities (Centar Sarajevo, Hadžići, Ilidža, Ilijaš, Novi Grad Sarajevo, Novo Sarajevo, Stari Grad Sarajevo, Trnovo i Vogošća). This study excluded seven private schools.

Firstly, we grouped the nine regions of Canton Sarajevo into three SES categories based on average family income, education and unemployment rate. We also took into account the number of inhabitants in the region, as regions with over 50,000 inhabitants have more schools.

We included in this study a total of four regions representing each SES category: High SES-Centar Sarajevo; Middle SES-Novo Sarajevo; Low-Novi Grad Sarajevo and Ilidža. After selecting the regions, we randomly selected 18 schools to represent Canton Sarajevo, allotted as follows: six schools each from both Centar and Ilidža, and three schools each from both Novo Sarajevo and Novi Grad.

### 2.2. Data Collection

From each of these schools, we randomly selected one third-grade class. The survey targeted a total of 441 child-parent pairs. Of these, 76 child-parent pairs were excluded because the parents did not want to participate. We excluded two child-parent pairs because their parents were illiterate. Thus, the study included 363 child-parent pairs, yielding a response rate of 82%. The percentage of boys and girls was 49% and 51%, respectively.

The survey asked pupils to voluntarily complete an anonymous self-administered questionnaire. The children received another questionnaire to take home to their parents to complete. After completing the survey, the pupils immediately returned it to the teacher or supervisor. During the study, the surveys used codes instead of names or surnames to match the parent/child questionnaires. Five children questionnaires (1.3%) were missing information about gender.

The surveys were based on earlier international research among school children, but contained minor modifications for the Bosnian circumstances [24,25].

### 2.3. The Child Questionnaire

The child questionnaire contained seven questions related to tooth brushing frequency, usage of fluoride toothpaste, tooth brushing habits and consumption of sweets with multiple answer options from which the respondent chose the one they found most suitable. The first question “How often do you brush your teeth?” offered five answer options: More than daily, daily, a couple of times a week, weekly, and rarely or never. We later dichotomised this question into two options: Twice daily (more than daily) or less than twice daily (all other answer options) according to the recommended toothbrushing behaviour [26]. The second question “Who brushes your teeth?” offered three answer options: I brush alone, my parents/guardian help me and my parents/guardian brush them. The third question “Do you use fluoride toothpaste while brushing?” offered four answer options: Always or almost always, often, sometimes and no. We later dichotomised the third question into two options: always or almost always, or less frequently (often, sometimes and no). The fourth question “How often do you see blood when brushing?” offered four answer options: Always or almost always, often, sometimes and no. The fifth question “How often do you drink juice/soda?” offered four answer options: More than once daily, daily, sometimes but not every day and rarely. The sixth question “How often do you eat sweets (cookies, cakes, chocolate)?” offered four answer options: More than once daily, daily, sometimes but not every day and rarely. The last question “Do you like and how often do you eat hard sweet treats (lollipops, hardy candy)?” offered five answer options: More than once daily, daily, sometimes but not every day, rarely and I don’t like those kinds of sweets.

### 2.4. Parent Questionnaire

The questionnaires for the parents also used multiple-choice questions from which the respondent circled the one answer they found most suitable. We divided the questions into four groups; the first group of questions related to the respondents’ oral hygiene habits. The first question “How often do you brush your teeth?” offered five answer options: More than once daily, daily, a couple of times a week, weekly and rarely or never; we later dichotomised them to twice daily or less than twice daily as for the children. The second question “What do you use to brush your teeth?” offered three answer options: toothbrush, dental floss and interdental toothbrush. The third question “Do you use fluoride toothpaste while brushing?” offered four answer options: Always or almost always, often, sometimes and no. The fourth question “How often do you drink juice/soda or coffee with sugar?” offered four answer options: More than once daily, daily, sometimes but not every day and rarely. The fifth question “How often do you eat sweets (cookies, cakes, chocolate)” offered four answer options: More than once daily, daily, sometimes but not every day and rarely. The sixth question “Do you like and how often do you eat hard sweet treats (lollipops, hard candy)?” offered five answer options: More than once daily, daily, sometimes but not every day, rarely and I don’t like those kinds of sweets.

The second group of questions related to the oral health habits of the respondent’s child and included four questions. The first question “How often are your child’s teeth brushed?” offered four answer options: twice daily, daily, a couple of times a week and sometimes. The second question “Who maintains oral hygiene in your child?” offered three answer options: the child alone, the child with the help of a parent/guardian, a parent/guardian maintains the child’s oral hygiene. The third question “How often do you oversee your child’s toothbrushing?” offered four answer options: Always or almost always, often, sometimes and rarely or never. We later dichotomised this into two options: Regularly (always or almost always, often) and rarely (sometimes and rarely, never). The final question in this group “How often do you forbid your child from consuming sweets and soda?” offered four answer options: Always or almost always, often, sometimes and rarely or never.

The third and fourth groups of questions related to the respondent’s attitude towards oral health/hygiene and oral health knowledge and included five answers options on a Likert scale: Completely agree, agree, disagree, completely disagree and I have no opinion. The questions were: “Teeth problems can cause other health issues”, “tooth diseases are less important than other diseases”, “it is normal that people lose teeth as they get older”, “baby teeth are not important because they fall out early”, “eating sweets and sugars causes tooth decay”, “microbial plaque causes caries”, “microbial plaque causes gum diseases”, “brushing twice daily with fluoride toothpaste will prevent diseases of the oral cavity”, “it is beneficial to visit the dentist regularly (every six months to a year)” and “rinsing with salt water or mouthwash is enough to keep the oral cavity clean”.

The last part of the questionnaire related to SES. The first question inquired about the highest level of education in the family and offered seven answer options: Illiterate, elementary school, high school, community college, college, masters, and doctorate. We later categorized them into three groups: Low (8 years of compulsory or illiterate), middle (vocational or technical training) and high (polytechnic or academic) education. The second question “Number of family members that live in your household?” offered four answer options: five or more, four, three and two. The third question “Which part of town do you live in?” offered four answer options: city centre, other neighbourhood in the city, suburbs and exurb. The fourth question “What is your marital status?” offered four answer options: married, divorced, single mother/father and widow; we later dichotomised them into two groups: Married and other. The fifth question “Are you currently employed?” offered four answer options: full time, part-time, unemployed and retired. For the analysis, we dichotomised these answers into two categories: full time and part time/unemployed. The sixth question “Is there a dental clinic available to you and your child?” offered four answer options: very near to us, not far from us, far from us and unavailable. The final question “Your previous experiences with the dentist?” offered five answer options: very positive, good, not so good, very negative and I don’t visit the dentist.

### 2.5. Statistical Analysis

For the statistical analysis, we used the Statistical Package for Social Sciences version 25.0 (IBM SPSS Statistics for Windows. IBM Corp., Armonk, NY, USA). Observation of cross-tabulations according to gender, tooth-brushing frequency, fluoride paste usage and consumption of sugary drinks and sweets with the chi-square test served in the statistical evaluation; *p* < 0.05 indicated statistical significance. Chi-squared tests also served to measure differences between the parents’ background characteristics: attitudes towards oral health/hygiene and oral health knowledge, employment, education, marital status, their own health behaviour, and their oral health behaviour towards their children. We used a multivariate logistic regression model to evaluate the probability of a child brushing their teeth according to recommendations and links to gender, region and parental employment; the parental question related to how often they brush their own teeth and how often they oversee their child’s tooth brushing.

## 3. Results

Table 1 presents the distribution of the children according to their school region, parents’ background and adherence to oral hygiene recommendations by gender. About one-third (34%) of the children came from schools in the Centre region and one-third (32%) from schools in the most peripheral parts of the town. More than half (58%) of the parents had reported high education, whereas 31% reported only basic education. The proportion of full-time employment was 78%. Two thirds (66.5%) of the children reported brushing their teeth twice daily (Table 1). Of the girls, 74% brushed their teeth twice daily; the corresponding figure for the boys was 58%.

Table 2 shows background information with regard to the socio-economic situation of the regions surveyed. The unemployment rate in each region shows the number of people not working but otherwise eligible. The high-SES region had a 14% unemployment rate, the middle-SES region 24% and the low-SES regions up to 40%. Further data showed differences in distribution between tertiary (TE), secondary (SE) and primary education (PE). The high- to low-SES regions showed an upward trend in differences in the unemployment rate between TE and PE education (Table 2), whereas SE remained consistent throughout. Gross domestic product and average salary accounted for the reported discrepancy between different SES regions; people working in higher SES regions had a higher gross domestic product and higher average salary than did their lower SES counterparts.

Table 3 shows tooth brushing frequency by background variables and oral health habits. Most (74%) of the girls reported brushing their teeth twice daily whereas about half (58%) of the boys brushed twice daily (*p* = 0.004). Children living in middle- and high-SES regions reported better oral health habits than their low-SES counterparts. The children of parents who reported twice-daily tooth brushing reported better oral health habits. A majority (69%) of the parents working full time had children who brushed twice daily, whereas 56% of the children with parents not working full time or unemployed brushed twice daily. Parents’ education was not significantly associated with children’s twice-daily tooth brushing.

In the logistic regression model (Table 4), boys and families living in the lower SES region were less likely to brush their teeth according to recommendations than were girls and those from middle- or high-SES regions. Parent’s own tooth brushing habit was significantly associated with their child’s twice-daily tooth brushing (OR: 3.45, CI 95%: 1.8–6.7; *p* = 0.000). Children of parents with a full-time occupation were more likely to brush their teeth according to the recommendation than those of unemployed parents, but the difference was not statistically significant (OR: 1.40, CI 95% 0.8–2.5).

## 4. Discussion

Our study showed alarmingly deficient oral health habits among schoolchildren in Sarajevo and point to a high risk for caries and other oral diseases in the future. These findings provide a basis for planning oral health promotion and prevention in Bosnia and Herzegovina where no public or subsidised dental health care services for children are presently available. Improving the oral health of children in the future will urgently require comprehensive oral health promotion and oral disease prevention programmes.

### 4.1. International Comparisons

Oral health habits among the Sarajevo children were poor; only two-thirds of nine-year-olds brushed their teeth twice daily and almost half of them failed to use fluoride toothpaste daily. In addition, the consumption of sugary products among the children was high. More than one-third of the children drank sugary drinks or beverages and almost half of them consumed sweets and sugary snacks daily or more. The consumption of sugary drinks and foods exceeded 10% of the WHO’s recommended total daily energy intake [4] and was higher than the consumption rate in e.g., Iceland, Norway and Germany [27]. Furthermore, almost 80% of the parents reported eating or drinking sugary foods daily or more, with the literature showing the association between parental and child consumption [8].

Children in Bosnia have one of the highest relative risks for caries in Europe [14]. Studies carried out among 6- to 12-year-olds in other similar high-risk countries show that Slovakia [28], Albania [29], Croatia [30] and Serbia [31] reported lower DMFT indices in children than did Bosnia, and an upwards trend in oral health. Further comparison to other European countries shows divergence. In the UK and the Nordic countries, for instance, DMFT values range from 0.4 to 1.4 [32,33,34]. These comparisons demonstrate a need for major improvements in Bosnia’s dental health system.

Our study showed a significant association between children’s twice-daily tooth brushing and their parents’ comparable oral health behaviour, which aligns with previous studies among younger children [7,9,10]. Parental oral health behaviour [7,9] directly influence the oral health habits of their children. A majority of parents are aware of their children’s oral health and their children often have better oral health status [11]. The gender difference in tooth brushing behaviour in favour of girls is in line with earlier studies [2,29].

Key risk factors for caries are lack of proper oral hygiene, lack of fluorides and excessive sugar consumption [6,12]. The role of parents is crucial in this; parents and professionals working with children should be aware of the sugar content of food and beverages and limit their children’s daily consumption [6]. Reducing exposure to risk factors such as high intake of free sugars at a young age decreases the negative and cumulative health effects later in life [4]. Educating the parents about the benefits of fluorides is also beneficial and could serve as a part of a future plan to improve oral health status in Bosnia.

Bosnian children coming from higher SES region were more likely to brush their teeth according to recommendations, which is in line with the results of other studies [2,3,23]. Our findings demonstrate that area of residence, parental employment and SES affect oral health behaviours. In Bosnia, the living area seems to be of greater importance than occupation and level of education with respect to oral health behaviour, indicating the societal structure and development in the country [19]. People with a higher education tend to have better-paying jobs and consequently live in better socio-economic conditions. The difference between urban and rural areas is large when it comes to caries prevalence, with children from rural areas showing worse dental health [3]. Over 40% of the study participants lived in low-SES regions and only 12% in high-SES areas. People with higher education levels often have higher oral health knowledge [22]. People with a lower education tend to live in rural areas and tend to be unsure about and even in disbelief about the benefits of regular tooth brushing. This study showed a significant relationship between parental employment and child oral health habits, with the children of parents working full time reporting better tooth brushing. This result shows worrying signs given the high (14–40%) unemployment rates in the area.

### 4.2. National Aspects

Bosnia and Herzegovina have seen no national oral health surveys conducted in the past 30 years. Previous regional or local studies in Canton Sarajevo and other parts of the Federation have shown that oral health among children in Bosnia and Herzegovina is poor. The 1992–95 war negatively impacted social conditions and the health care system, which explains the poor dental health among the children there. In 2004, the mean DMFT index in 6 and 12- to 15-year-olds varied between 4.2 and 7.6 [21]. Later research in the same age groups showed DMFT indices of 2.6, 5.5, and 6.2, respectively [31,35]. A more recent study showed no improvement in these indices [18].

Activities promoting oral health in Bosnia and Herzegovina have not benefitted from profound and systematic planning, monitoring or implementation at a national level. On the contrary, such activities have involved only short-term projects on a local or regional level. Unsurprisingly, the results have seen little success. Several federal agencies carried out a health promotion and prevention programme with the primary objective to familiarise first-grade children with the importance of proper oral hygiene [36]. However, although the Federal Ministry of Health has declared oral health to be nationally important, the Ministry has failed to provide a clear strategy for the activities. The drinking water in Bosnia and Herzegovina has very low levels of natural fluorides, with no water fluoridation [21].

Eastern European countries that have established and maintained school oral health programmes have seen improved dental health among the children there [34]. The WHO has declared that universal access to fluoride for caries prevention must be a part of the basic human right to health [37]. The WHO goal for 2020 is that at least 80% of 6-year-olds be caries-free and no more than 1.5 DMFT among 12-year-olds [34]. Eastern Europe will achieve better oral health goals only if they implement oral health promotion and oral disease prevention programmes in schools according to the recommendations of the WHO Health Promoting Schools Project [34]. To achieve this, three elements are crucial: health education, primary and secondary prevention [12]. Oral prevention programmes for children are a necessity to reduce the prevalence of caries in school-age children [12].

Based on these results, the objectives, directed to the children as the target population for planned preventive programmes, should focus on those that can be achieved cheaply and quickly, such as using fluoride rinses in kindergartens and schools, controlling children’s diet and snacks during school time or handing out fluoride toothpaste to children. Long-term objectives should focus on establishing a public health programme that is closely connected to kindergartens and schools, establishing schools for dental hygienists, centralising oral health services with mandatory regular check-ups and fluoridation of the drinking water.

### 4.3. Strengths and Limitations

This study sheds light on the largely unknown current oral health situation among schoolchildren and their families in Bosnia and Herzegovina. We applied the cluster sampling method to obtain a representative sample from the largest centre in the Federation. This study has also limitations. Due to its cross-sectional design, this study proposed no causal interpretations. Self-reporting surveys may be susceptible to socially desirable answering, but because the participants answered voluntarily and anonymously, so the results can be considered reliable.

## 5. Conclusions

Our study showed that Sarajevo children’s oral health habits were poor. One-third of the nine-year-olds failed to brush their teeth according to recommendations, and almost half of them failed to use fluoride toothpaste daily. Improving the children’s oral health in the future will urgently require national oral health promotion and prevention programs.

## Figures and Tables

**Table 1 ijerph-18-03235-t001:** Distribution of the children according to their school region, parents’ background and adherence to oral hygiene recommendations by gender.

	Boys	Girls	All
Variable	N = 175 (%)	N = 183 (%)	N = 358 (%)
**Region**			
Centar	61 (34.9)	62 (33.9)	123 (34.4)
Novo Sarajevo	24 (13.7)	29 (15.8)	53 (14.8)
Novi Grad	33 (18.9)	34 (18.6)	67 (18.7)
Ilidza	57 (32.6)	58 (31.7)	115 (32.1)
**Parents Education**			
High	102 (59.6)	105 (57.7)	207 (58.6)
Middle	16 (9.4)	22 (12.1)	38 (10.8)
Low	53 (31)	55 (30.2)	108 (30.6)
**Parents employment**			
Full time	135 (77.1)	145 (79.2)	280 (78.2)
Part-time/unemployed	40 (22.9)	38 (20.8)	78 (21.8)
**Marital status**			
Married	157 (90.8)	170 (93.9)	327 (92.4)
Others	16 (9.2)	11 (6.1)	27 (7.6)
**Tooth brushing frequency**			
Twice daily	102(58.3)	136 (74.3)	238 (66.5)
<Twice daily	73 (41.7)	47 (25.7)	120 (33.5)

**Table 2 ijerph-18-03235-t002:** General information about the regions in Canton Sarajevo according to national statistics *.

	Centar	Novo Sarajevo	Novi Grad	Ilidza	City Total
Variable	N (%)	N (%)	N (%)	N (%)	N
No. of residents	54,091 (12.9)	64,548 (15.4)	120,314 (29)	69,533 (16.6)	418,542
Age ^#^					
<14	10,333 (14.8)	10,140 (13.8)	20,208 (16.2)	11,632 (19.6)	
15–64	46,574 (66.6)	48,611 (66.2)	86,154 (69.1)	39,030 (65.9)	
>65	12,982 (18.6)	14,643 (20)	18,380 (14.7)	8609 (14.5)	
Unemployment rate	7171	7515	16,873	10,520	
Unemployed with TE	1650 (23)	1691 (22.5)	2468 (14.6)	1340 (12.7)	
Unemployed with SE	3948 (55)	4358 (58)	9925 (58.8)	6081 (57.8)	
Unemployed with PE	1573 (22)	1466 (19.5)	4480 (26.6)	3099 (29.5)	
Unemployment rate (%)	14.3%	24%	39.8%	33.5%	
Gross domestic product	33,309	20,211	7198	7130	
Average salary (% of city av)	1217 (+32)	1176 (+27)	917 (−0.5)	876 (−5)	921
Children starting schools/year	5078 (13.6)	5306 (14.3)	10,251 (27.6)	6858 (18.4)	37,077

* Agency for statistics of Bosnia and Herzegovina, 2018. ^#^ Data from 2012.

**Table 3 ijerph-18-03235-t003:** The distribution of tooth brushing frequency by gender, fluoride toothpaste usage, and factors related to the area of residency and parental attitude and background.

	<Daily N (%)	Daily N (%)	Twice Daily N (%)	Total N (%)	*p*-Value
**Gender**					0.004
Boy	35 (64.8)	38 (57.6)	102 (42.9)	175 (48.9)	
Girl	19 (35.2)	28 (42.4)	136 (57.1)	183 (51.1)	
**Region**					0.006
Centar	8 (14.8)	20 (29.9)	96 (39.7)	124 (34.2)	
Novo Sarajevo	8 (14.8)	7 (10.4)	42 (17.4)	57 (15.7)	
Novi Grad	14 (25.9)	14 (20.9)	39 (16.1)	67 (18.5)	
Ilidza	24 (44.4)	26 (38.8)	65 (26.9)	115 (31.7)	
**Child Fluoride-Toothpaste Usage**					0.007
Always	14 (25.9)	29 (43.3)	128 (53.3)	171 (47.4)	
Often	13 (24.1)	16 (23.9)	40 (16.7)	69 (19.1)	
Sometimes	15 (27.8)	16 (23.9)	50 (20.8)	81 (22.4)	
No	12 (22.2)	6 (9)	22 (9.2)	40 (11.1)	
**Parents Tooth Brushing**					0.000
<Daily	8 (14.8)	3 (4.5)	6 (2.5)	17 (4.7)	
Daily	5 (9.3)	14 (20.9)	14 (5.8)	33 (9.1)	
Twice daily	41 (75.9)	50 (74.6)	222 (91.7)	313 (86.2)	
**Parents Employment**					0.032
Full time	35 (66)	51 (77.3)	198 (82.2)	284 (78.9)	
Part-time/unemployed	18 (34)	15 (22.7)	43 (17.8)	76 (21.1)	

**Table 4 ijerph-18-03235-t004:** Twice daily tooth brushing among the 9-year-olds by gender, school region, parental occupation, parent’s involvement in child’s tooth-brushing, and parental tooth-brushing by means of a binary logistic regression model.

Toothbrushing	S.E. *	*P*	OR	CI 95%
**Gender**				
Male			1	
Female	0.241	0.004	1.994	1.2–3.2
**Region**				
Low SES			1	
High/middle SES	0.249	0.001	2.294	1.4–3.7
**Occupation**				
Part-time/unemployed			1	
Full time	0.289	0.249	1.395	0.8–2.5
**How Often do You Oversee Your Child’s Toothbrushing?**				
Rarely			1	
Regularly	0.254	0.353	0.790	0.5–1.3
**Parent’s Own Tooth Brushing Habits**				
<Twice daily			1	
Twice daily	0.339	0.000	3.445	1.8–6.7
Constant	0.250	0.000		

* Standard Error (S.E.), *p*-value (*P*), Odds Ratio (OR), Confidence Interval (CI).

## Data Availability

The data presented in this study are available from the corresponding author on reasonable request.

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
