# Peer review of "Oral Health Behaviour of Nine-Year-Old Children and Their Parents in Sarajevo"

_ijerph, 2021, doi:10.3390/ijerph18063235_

Round 1

Reviewer 1 Report

The article appears interesting but some minor issue should be argued. 

The author described that the questionnaire has previously more option and then was later dichotomised, the A.A. should discuss why in the discussion section. 

Authors should improve the DMFT index related information, as done in this paper. Giudice GL, Cicciù M, Polimeni A, Lizio A, Giudice RL, Lauritano F, et al. Oral and dental health of Italian drug addicted in methadone treatment. Oral Sci Int 2019;16(1):8-14.

Authors should improve the discussion comparing the DMFT of geographically close region and high risk region, to let the reader to compare the data. 

Author Response

Authors should improve the DMFT index related information, as done in this paper. Giudice GL, Cicciù M, Polimeni A, Lizio A, Giudice RL, Lauritano F, et al. Oral and dental health of Italian drug addicted in methadone treatment. Oral Sci Int 2019;16(1):8-14.

We have amended the DMFT related information in the Introduction (p. 2, lines 54-57) and Discussion (p. 10, lines 254-255; p. 11, lines 290-292). Since our survey did not collect DMFT-data, a table would not be suitable.

Authors should improve the discussion comparing the DMFT of geographically close region and high risk region, to let the reader to compare the data. 

We have compared the DMFT-situation with countries in the region (Slovakia, Albania, Croatia and Serbia) (p. 10, lines 251-254). We omitted some of the geographically more distant countries.

Reviewer 2 Report

The authors have properly modified the purpose of this study according to my peer-reviewed comments. I think this manuscript is suitable for publication to the IJERPH.

Author Response

The authors have properly modified the purpose of this study according to my peer-reviewed comments. I think this manuscript is suitable for publication to the IJERPH.

We thank the reviewer for the comments and suggestions.

Reviewer 3 Report

This is a revised version of a previously submitted paper.  The authors have made the changes suggested and the result is a much improved article that merits publication.

Author Response

This is a revised version of a previously submitted paper.  The authors have made the changes suggested and the result is a much improved article that merits publication.

We thank the reviewer for the comments and suggestions.

Reviewer 4 Report

Thank you for asking me to review this paper on 'Oral Health Behaviour of Nine-Year-Old Children and Their Parents in Sarajevo'.

The manuscript is well written and sheds further light into a urgent public health issue - focusing on an area of high dental decay. There are some recommendations I have for this paper detailed below.

Page 3 - you state there were 441 children and 363 child parent pairs, could you be clear about the distribution so i assume x number of pairs with one child and parent, x number with 2 children and parent? (how many twins, triplets etc)

Page 3/4 the section detailing the child and parent questionnaire is quite long. I do think you should provide this information - I wonder if it would be better as a table - detailing the question, responses and then also if these were dichotomised for  analysis to make it easier for the reader. This is just a suggestion and not necessary if the authors prefer not.

The analysis concentrated on brushing twice a day or not. Did you carry out any similar analysis looking at the use of fluoridated toothpaste (are the same factors associated with this).Why brushing was the factor looked at in this way. For the future it would be important to both increase brushing frequency and the use of fluoridated toothpaste? Was there any data collected on dental attendance?

Can you confirm - there was no signed consent form as you assumed that by filling in the questionnaire they were providing consent? Were parents / children provided with any information sheet about the data collection etc?

Author Response

Page 3 - you state there were 441 children and 363 child parent pairs, could you be clear about the distribution so i assume x number of pairs with one child and parent, x number with 2 children and parent? (how many twins, triplets etc)

We have added more information regarding child-parent pairs to clarify the numbers as suggested by the reviewer (p. 3, lines 96-97). Some of the parents did not want to participate.

Page 3/4 the section detailing the child and parent questionnaire is quite long. I do think you should provide this information - I wonder if it would be better as a table - detailing the question, responses and then also if these were dichotomised for analysis to make it easier for the reader. This is just a suggestion and not necessary if the authors prefer not.

We thank for this comment. This section is quite long due to the fact that previous reviewers requested us to provide more details related to the variables and the dichotomisation.

The analysis concentrated on brushing twice a day or not. Did you carry out any similar analysis looking at the use of fluoridated toothpaste (are the same factors associated with this). Why brushing was the factor looked at in this way. For the future it would be important to both increase brushing frequency and the use of fluoridated toothpaste? Was there any data collected on dental attendance?

We analysed use of fluoridated toothpaste: use of F-toothpaste was associated with tooth brushing frequency among the children as shown in Table 3 (p. 8). In the regression model twice-daily tooth brushing was found to be significant. It is true that to increase tooth brushing frequency and use of F-toothpaste are important and we have highlighted this in the Discussion (pp. 9-10) (lines 243-244, 268-270). We did not collect data of dental attendance.

Can you confirm - there was no signed consent form as you assumed that by filling in the questionnaire they were providing consent? Were parents / children provided with any information sheet about the data collection etc?

We provided detailed written information about the study to all parents (p. 3). The study followed the guidelines of the Helsinki Declaration, and was voluntary and anonymous. The children were given oral information at school about the study and were asked to participate voluntarily.

The author described that the questionnaire has previously more option and then was later dichotomised, the A.A. should discuss why in the discussion section.

WHO recommendation was used as criteria for dichotomisation (p. 4, lines 112-114) as requested by one reviewer. F-toothpaste usage was also dichotomised according to recommendations. Other variables (regions by SES, marital status, employment, and parental control) were dichotomised for the logistic regression analyses. We discussed these and compared the findings with other studies on page 2 (lines 44-46) and on page 10 (lines 261-262, 272-275).

Round 2

Reviewer 4 Report

The authors have addressed the main concern from the review

This manuscript is a resubmission of an earlier submission. The following is a list of the peer review reports and author responses from that submission.

Round 1

Reviewer 1 Report

The article Oral health behavior of nine-year-old children and their parents in Sarajevo is a correctly written cross-sectional study article. The methodology and results were correctly described. However, the topic is not innovative and not at the level of this journal- in my opinion articles based on a patient survey should not be published in a highly impacted journal. References are too old - there should be no articles older than 10 years. The conclusions are not surprising, they actually repeat what is already known and described in the introduction.

1. The article does not have enough novelty characteristics. The prepared questionnaire covers a narrow scope and is specific to a narrow group of respondents, including only Sarajevo. The conclusions are confirmed by facts known for years in scientific publications.

2. The range of questions is too wide and concerns two matters at the same time, e.g. 'Do you like and how often do you eat hard sweet treats (lollipops, hardy 131 candy)? how to answer unequivocally yes or no.

3. The order of the questions is not consistent and logical. Question 4 'How often do you drink 125 juice / soda?' 5 'How often do you see blood when brushing?' Question 6 'How often do you eat sweets (cookies, cakes, chocolate)?' The sequence of questions is not well thought out. In between the diet questions, there's a question about blood.

Reviewer 2 Report

The article is interesting and gives a good overview on the present oral health situation of this region. 

Some aspects must be improved 

Introduction 

The introduction should be integrated with more data retrieved from the who guidelines for sugar intakes and oral health habits. 

This aspect should also be compared and discussed in the discussion section. 

Method

Did you performed a power analysis ?

What does the Authors means when says "We later dichotomised the answers"? A.A. should describe the method in text and report the data accordingly.

The A.A. should expand the discussion section letting the readers understand which data have a statistical significance.

The A.A. should discuss the results found in this analysis 

It could interesting for the reader to evaluate if there is a statistically evident correlation between the parents education and the tooth brushing frequency and if this data is correlated with the child brushing frequency

The data should reformatted dividing the demographic data, the data retrieved from the child questionnaire, and the data retrieved from the parents questionnaire. 

An additional tab could be made to let the reader understand which parameters have been statistically correlated, this aspect should be underlined in text

Some of the question reported in the questionnaire are not related to any data in any tab. 

Reviewer 3 Report

The purpose of this study was to examine the association between the oral health behavior and oral health knowledge of the parents and the oral health habits of their children. The researchers predicted that parents with better oral health behavior and higher oral health knowledge who live in better socio-economic conditions would have children with better oral health habits. However, some studies have already shown that family's socio-economic status (SES) critically influences dental caries with a clear correlation between low SES and children’s oral health.(Lines 66-72) So, I could not understand the novelty of this study. I think the purpose of this study is to investigate the oral health behavior in Bosnia and Herzegovina.

Reviewer 4 Report

This is an extremely well written paper.  It reports an important study of the oral health of children aged 9 years in Sarajevo according to their socio-economic status.  Findings show that children are at high risk of caries due to their generally poor level of oral health care.  The study found that around a third of all children fail to brush their teeth daily, and that fluoride toothpastes were used much less than is desirable.  The authors rightly point out that these factors will have to change if the Government's oral health targets are to be met.  This is therefore an important study, and one that should be published.

However, there are two minor concerns about the manuscript.  One is the repeated use of the word "parent" on its own (lines 85, 103, 135, 286) whereas elsewhere, the term "parent/guardian" is used (lines 121, 155).  The latter term implies that there were some family units where the children were cared for by someone other than their biological parent, and expressing the family situation in this way is to be preferred in studies of this type.

Secondly, in line 268, the abbreviation DMFT/dmft is used.  Since the authors have defined DMFT (with capital letters) earlier in the paper, using both capital and lower case letters is not necessary, and it should be expressed simply as "DMFT".

Subject to making these minor corrections, the paper is suitable to proceed to publication.